# Effect of Silica Sodalite Loading on SOD/PSF Membranes during Treatment of Phenol-Containing Wastewater

**DOI:** 10.3390/membranes12080800

**Published:** 2022-08-19

**Authors:** Olawumi O. Sadare, Rivoningo Ngobeni, Michael O. Daramola

**Affiliations:** 1Department of Chemical Engineering, Faculty of Engineering, Built Environment and Information Technology, University of Pretoria, Hatfield, Pretoria 0028, South Africa; 2School of Chemical and Metallurgical Engineering, Faculty of Engineering and the Built Environment, University of the Witwatersrand, Private Bag X3, Wits, Johannesburg 2050, South Africa

**Keywords:** polysulfone, phenol-containing water, polymer matrix, mixed matrix membrane, wastewater treatment, silica sodalite

## Abstract

In this study, silica sodalite (SSOD) was prepared via topotactic conversion and different silica sodalite loadings were infused into the polysulfone (PSF) for application in phenol-containing water treatment. The composite membranes were fabricated through the phase inversion technique. Physicochemical characteristics of the nanoparticles and membranes were checked using a Scanning Electron Microscope (SEM), Brunauer Emmett–Teller (BET), and Fourier Transform Infrared (FTIR) for surface morphology, textural properties, and surface chemistry, respectively. A nanotensile test, Atomic Force Microscopy (AFM), and contact angle measurement were used to check the mechanical properties, surface roughness, and hydrophilicity of the membranes, respectively. SEM results revealed that the pure polysulfone surface is highly porous with large evident pores. However, the pores decreased with increasing SSOD loading. The performance of the fabricated membranes was evaluated using a dead-end filtration device at varying feed pressure during phenol-containing water treatment. The concentration of phenol in water used in this study was 20 mg/L. The pure PSF displayed the maximum phenol rejection of 95 55% at 4 bar, compared to the composite membranes having 61.35% and 64.75% phenol rejection for 5 wt.% SSOD loading and 10 wt.% SSOD loading, respectively. In this study, a novel Psf-infused SSOD membrane was successfully fabricated for the treatment of synthetic phenol-containing water to alleviate the challenges associated with it.

## 1. Introduction

Over 1.1 billion people lack access to sufficient potable water supply; this is a result of the high costs of potable water, rapid increase in population, and environmental and climatic factors [1]. The number of people affected by water shortages has been predicted by the World Health Organization to reach up to 4 billion by 2050 [2]. Water constitutes more than 70% of the earth’s surface. However, 96% of it contains too much salt for human consumption, and the remaining 3% is trapped under the ground, ice caps, and glaciers. The remaining less than 1% of the world’s water supply is accessible for human use [3]. In addition, human activities such as industrial developments are threatening the security of the water resources accessible for human consumption and domestic purposes [4]. Different organic pollutants play major roles in water pollution. The prevalent organic pollutants in industrial wastewater include phenols, aliphatic, heterocyclic compounds, and polycyclic aromatic hydrocarbons [5].

Phenolic compounds result in severe environmental issues in effluents due to their poor biodegradability and high toxicity [6], resulting in the pollution of potable water, plants, and sea animals [7,8]. They are usually found in wastewaters released by various manufacturing processes, for instance, in the petrochemical industry, oil refineries, synthetic rubber production, plastics, cooking, phenolic resin industries, paper, and ceramic production at varied concentrations [9,10]. Phenols have been categorized as harmful pollutants and they are given priority as they are lethal to organisms and humans even at minimum concentrations [9]. They have therefore been subjected to specific regulations owing to their high toxicity, and their industrial application has been substituted with less harmful compounds [11]. The recommended allowable phenolic concentration in drinking water according to World Health Organization is 1 µg L^−1^, and less than 1 µg L^−1^ concentration in wastewaters as regulated by the European Union and the US Environmental Protection Agency (EPA) [12]. Furthermore, phenol has been identified to effortlessly penetrate the human body through the respiratory tract and the skin. Therefore, numerous efforts have been put into developing approaches that could eradicate or reduce the release of phenol into the water stream. The technologies include adsorption, photocatalysis, membrane separation, and precipitation to mention but a few [13]. Activated carbon with high adsorption capacity has been widely used as an adsorbent for the effective treatment of various organic pollutants. Compared to the adsorption technique, which is a physical, rate-controlled, and equilibrium-driven process, separation by the membrane is quicker and more effective. Moreover, orthodox membrane technology is dependent on molecular sieving. The application of compact membranes during organic compounds treatment makes the process energy-demanding. Different membrane separation processes have been used, such as reverse osmosis [14], nanofiltration [15], and pervaporation membranes [16]. The major drawbacks of these technologies include high energy demand and the inability to efficiently remove trace amounts of solute.

Furthermore, a number of investigators have utilized the application of activated carbon and then membrane filtration in a hybrid process [17,18]. This hybrid process requires more than one stage of applications, thereby making the process more complex, laborious, and difficult to maintain. Nevertheless, the current ultra-modern membrane science requires the operation of dualistic properties, in the form of a mixed matrix membrane (MMM). An MMM works such that it adsorbs smaller solutes and filters the bigger solutes in a single stage. Through this method, high removal selectivity is achievable using a more open membrane. Therefore, the application of MMM in the selective removal of solute with high efficiency at reduced feed pressure has received a lot of attention recently [13,19]. 

Ntshangase et al. [20] studied the performance of functionalized silica sodalite (fSSOD)-infused Polysulfone (PSF) membranes for the effective removal of heavy metals in acid mine drainage (AMD). The 10% fSSOD/PSF displayed about >50% rejection for all the heavy metals investigated at 4 bar. Likewise, an investigation was conducted recently by Ngobeni et al. [21], where SSOD and hydroxy sodalite (HSOD) crystals were incorporated into PSF to improve the physico-chemical property of the polymer for treating phenol-containing wastewater, and the rejection performances of these materials were compared. Results showed a 64.75% phenol rejection by the 10 wt.% SSOD/PSF membrane. It has been observed from previous studies that varying the concentration of nanoparticles in composite membranes affects the permeability, selectivity, and mechanical property of the membrane. Varying the nanoparticle loading in a polymer matrix could change the physical and chemical characteristics of the membranes. Furthermore, the separation performance and integrity of the membranes with different silica sodalite loading may differ significantly. During an investigation carried out by Daramola et al. [22], it was observed that increasing the sodalite concentration in the PES matrix resulted in higher selectivity during AMD treatment. However, permeability decreased with increasing sodalite concentration. In another study conducted by Mukherjee and De [23], it was found that increasing the concentration of CNTs in a polysulfone matrix for the removal of toluene, benzene, and phenol reached the optimum selectivity at 4 wt.%. Membranes with a CNT concentration higher than 4 wt.% did not show further improvement in the selectivity of the composite membrane [23]. 

As far as it can be established, it is noteworthy that limited research has been conducted and reported in the literature on the effects of SSOD nanoparticles loading in mixed matrix membranes for use in phenol-containing wastewater treatment. In view of this, this current study explores the effect of silica sodalite loading on the polysulfone matrix membrane during phenol-containing wastewater filtration.

## 2. Materials and Methods

Phenol (99%; 94.11 g/mol), Beaded polysulfone (22,000 g/mol), and *N,N*-dimethylacetamide (99%) were procured from Merck, Johannesburg, South Africa. SSOD crystals were prepared by topotactic conversion, using Tetra-ethoxysilane (Purity, 99%) and tetramethylammonium hydroxide (Purity, ≥97%), both purchased from Merck, Johannesburg, South Africa. The Nitrogen gas was procured from AFROX, Johannesburg, South Africa. All the chemicals purchased for this study were utilized without any additional purification.

### 2.1. Preparation of Silica Sodalite (SSOD) Crystals

Synthesis of silica sodalite nanoparticles was performed via topotactic conversion by an established technique developed by Moteki et al. [24]. In a Teflon cup, about 18.52 g (25 wt.%) of tetraethoxysilane and 32 mL of an aqueous solution of tetramethylammonium hydroxide was gently added. To ensure homogeneity, the mixture was agitated for 24 h on a magnetic stirrer. The solution was poured into a Teflon-lined stainless-steel autoclave and set at 413 K for 7 days. The SSOD crystals were sieved and rinsed with acetone, after which the solid particles were oven-dried at a temperature of 333 K for 24 h. 

### 2.2. Fabrication and Characterization of Sodalite-Infused Polymer Membranes

#### 2.2.1. Fabrication of SSOD/PSF Membranes

Acid pre-treatment was achieved by dissolving 0.1 g of RUB-15 in 30 mL of 5 M propionic acid. The solution was agitated at ambient temperature for 3 h. The solid particles were sieved and rinsed with deionized water and oven-dried for 24 h at 333 K, after which the solid particles were calcined at 1073 K for 5 h. In this study, the fabrication of the asymmetric membranes was performed through the phase inversion method as reported by Gohil and Ray [25]. Two different Silica sodalite nanoparticles loadings (5 wt.% and 10 wt.%) were poured into 50 mL of *N*,*N*-dimethylacetamide. The mixture was ultra-sonicated for 15 min and stirred for additional 15 min. To the mixture of SSOD and the *N,N*-dimethylacetamide, about 10 g of PSF was added, ultra-sonicated for 15 min, and agitated at 400 rpm on a magnetic stirrer for 24 h. The solution was poured on a smooth glass plate with the help a casting blade. The cast solution was allowed to remain on the glass plate for about 10 s and subsequently the glass plate with the cast solution on it was completely dipped in deionized water for 24 h. The fabricated membranes were oven-dried for 20 min at 60 °C, in order to get rid of the solvent. The dried fabricated membranes were then kept for further physico-chemical characterizations. 

#### 2.2.2. Characterization of SSOD Crystals and Fabricated SSOD-Infused Membranes

An X-ray diffractometer (XRD) (model Bruker D2 Phaser) was used to check the crystallinity of the SSOD particles. The pore size and the surface area of SSOD crystals were examined using the Brunauer–Emmett–Teller (BET) analysis. A Scanning Electron Microscope (SEM) produces high-resolution 3D images by scanning a material-focused beam of electrons. The surface structure of the prepared SSOD particles and the surface of the fabricated membrane were coated with gold/palladium and examined with an SEM. However, to study the membranes’ cross-section, membranes were immersed in liquid nitrogen for 1 min, and immediately after immersion cut open with razor blades to expose the cross-sectional area. Gold-palladium was used to coat the cross-sectional area of the fabricated membranes, and SEM images were taken.

Attachment of surface functional groups to the nanoparticles and fabricated membranes was studied with FTIR spectroscopy using PerkinElmer spectrum two. The nanoparticles and the synthesized membranes were placed on the diamond surface. The spectra were obtained in the range 400–4000 cm^−1^.

The surface roughness was determined using an atomic force microscope (AFM) Vi300 model as shown in Figure 6. The tendency of the membranes to foul during operation could be revealed by the roughness values [26].

The mechanical properties of the fabricated membranes were studied by means of a texture analyzer (Model: TA.XT.plus). To evaluate the highest force the fabricated membranes can resist prior to ripping, nanotensile tests were conducted and studied the membrane’s capacity to withstand elastic distortion under pressure [27].

Contact angle measurements were taken with the aid of dataphysics OCA 15 model, to determine the membrane wettability. The contact angle is the angle created with the solid when a liquid is deposited on it [28]. A micro-syringe is used for the measurement of contact angle, whereby drops of water are placed on the outer surface of the membrane, at 10 different spots. The mean value for each membrane sample was recorded and used as the surface hydrophilicity of the membranes [27]. To examine how much water a membrane can absorb, the equilibrium water content (EWC) of polymer membranes is usually analyzed. EWC is directly associated with porosity and can be regarded as the level at which the membrane neither loses nor gains moisture [29]. EWC is calculated using Equation (1)
(1)$$EWC=\frac{W_{w}-W_{d}}{W_{w}}\times 100\%$$
where **W_w_** is the wet weight of the membrane and **W_d_** is the dry weight of the membrane. The porosity of the membrane measures up the void spaces created by pores within the membrane, and can be calculated by the mass loss of the wet membrane after drying up using Equation (2) [19].
(2)$$Porosity=\frac{W_{w}-W_{d}}{\rho .V}\times 100\%\;$$
where ρ stands for the density of water and **V** is the membrane total volume.

### 2.3. Preparation of Phenol-Containing Wastewater and Its Treatment Using Fabricated Membranes

Phenol crystals amounting to 1 g were dissolved in 1 L of distilled water, the solution was thoroughly mixed together by stirring. Twenty milligrams of the stock solution were mixed with 980 mL of distilled water in a 1 L calibrated conical flask, to prepare 20 mg/L of stock solution. The evaluation of membrane performances was done using a dead-end filtration device. The device consisted of an inlet pipe for inert gas, a filtration cell for loading the feed, a pressure gauge, a magnetic stirrer plate, and a magnetic bar.

The synthesized membranes were fitted in the filtration cell one at the time. The deionized water was used as pure water which was set as the standard for comparison purposes. The filtration of pure water through membranes was used as the original flux for the monitoring of fouling activities [30]. Membrane was inserted in the cell, pure water was poured as the feed, and pressure was increased to 4 bar in order to equilibrate the system. This enables easy passage for water as membrane pores open [31]. In this study synthetic wastewater was used whereby the concentration of phenol in water was kept constant at 20 mg/L. The nitrogen gas from the inlet gas pipe was employed to exert varied pressure (4 to 7 bar) into the filtration cell. The homogenous solution of the diluted stock solution was maintained by constant stirring using a magnetic stirrer plate. The permeate samples from the filtration cell were taken for analysis. The untreated synthetic wastewater and permeate samples were analyzed using a pre-calibrated HPLC, Agilent 1200 series model, with Eclipse XDB C-18 column. The mobile phase used was acetonitrile/deionized water (3:1), the injection volume was 10 µL at a flow rate of 1 mL/min. The pure water and phenol-containing permeate fluxes were calculated using Equation (3). The percentage rejection (selectivity) of the membrane was obtained using Equation (4): (3)$$J=\frac{V}{A}$$
**J** is permeation flux (L/m^2^h); ***V***, the volumetric flow (L/h); and ***A***, the effective membrane area (m^2^).
(4)$$R=\frac{C_{F-}C_{P}}{C_{F}}\times 100$$
**R** is percentage rejection; ***C_F_*** is the concentration of phenol in the feed (mg/L); and ***C_P_*** is the concentration of phenol in the permeate stream (mg/L) [32].

Maximum average roughness (***R_a_***) and root mean square roughness (***R_qms_***) of the membrane samples were obtained from Equations (5) and (6), respectively [33]:(5)$$R_{a}=\frac{1}{L}\int_{O}^{L}IZ\left(x\right)Id_{x}$$
(6)$$R_{qms}=\sqrt{\frac{1}{L}\int_{O}^{L}IZ^{2}\left(x\right)Id_{x}}$$
where ***L*** is the length of the membrane sample at position (X), ***Z*** is the function of the surface profile of the sample under investigation

Cleaning techniques have to be applied to reverse the fouling in case the fouling is reversible, so as to increase the life expectancy of a membrane [20]. A Flux recovery experiment was conducted by cleaning the surface of the pure Psf membrane and SSOD-loaded membranes with deionized water for 30 min at 4 bar. The flux recovery ratio (FRR), which shows the membranes’ capability to recover from fouling, was calculated using Equation (7). A higher FRR signifies an improved antifouling property [20]. In this case, restoring the membrane flux after cleaning is much easier whereby there is a low reduction in flux over time. Fouling is classified as organic, inorganic, or biofouling which may be reversible or irreversible depending on the type of foulants [20]. To evaluate the antifouling property of membranes, total flux decline ratio (***R_t_***), reversible flux decline ratio (***R_r_***), and irreversible flux decline ratio (***R_ir_***) were evaluated, using Equations (8)–(10) [34]:(7)$$FRR\left(\%\right)=\frac{J_{w2}}{J_{w1}}\times 100$$
(8)$$R_{t}\left(\%\right)=\frac{J_{w1}-J_{P}}{J_{w1}}\times 100$$
(9)$$R_{r}\left(\%\right)=\frac{J_{w2}-J_{P}}{J_{w1}}\times 100$$
(10)$$R_{ir}\left(\%\right)=\frac{J_{w1}-J_{w2}}{J_{w1}}\times 100=R_{t}-R_{r}\;$$
where ***J_w_***_1,_
***J_w_***_2_ and ***J_p_*** are the initial flux, flux after a cleaning process, and the permeate flux of phenol during filtration process, respectively. 

## 3. Results

### 3.1. Physico-Chemical Characterization of SSOD Crystals

EWC and the porosity of the fabricated membranes are presented in Table 1. EWC is an essential parameter for characterization to indicate membranes’ hydrophilicity and hydrophobicity. Similarly, it is related to the pores, and to the total volume of the porous membrane. Membrane pores offer the possibility for the passage of water molecules. The higher the porosity, the higher the water content of the membrane [29]. The result in Table 1 signifies that the pure Psf membrane (0 wt.% SSOD/Psf) has the lowest EWC and porosity compared to the SSOD-loaded membranes. This is an indication that pure Psf is hydrophobic in nature. However, an increase in EWC and porosity was noticed when the membranes were loaded with SSOD nanoparticles. Furthermore, the EWC of the fabricated membranes increased from 64.05% to 73.24% when the SSOD loading increased from 5 wt.% to 10 wt.%. The porosity of the fabricated membrane was observed to increase from 55.40% to 61.09% as the SSOD nanoparticles increased from 5 wt.% to 10 wt.%. This could help to enhance the flux and permeability of the membranes during phenol removal. The results showed that SSOD loading increased the water content of the fabricated membranes. This is an indication of the availability of pores for easy passage of water. 

The textural properties of the SSOD crystals synthesized via topotactic conversion are presented in this study. The results show that SSOD has a specific surface area of 27.29 m^2^/g as compared to the surface area of HSOD (2.35 m^2^/g) synthesized via the hydrothermal method in our previous study [21]. This is an indication that synthesis through topotactic conversion resulted in a considerably improved specific surface area. The higher specific surface area suggests improved adsorption capacity, and therefore more pollutants from wastewater would be easily trapped onto the SSOD surface [35,36]. SSOD also has a larger pore volume of 0.067 (cm^3^/g) compared to 0.012 reported for HSOD by Ngobeni et al. [21]. This indicates that SSOD may have more accessible pores compared to HSOD. If incorporated into a polymer membrane, this could result in higher permeation flux during membrane performance evaluation as reported by Moteki et al. [24].

Figure 1 depicts the XRD patterns for SSOD synthesized for this study. It can be seen that SSOD crystals were successfully synthesized, having a matched pattern with the reference SSOD patterns obtained from the IZA website [37]. However, it can be observed that the intensity of the peaks for the synthesized SSOD is relatively lower when compared to the reference XRD patterns. This could be as a result of the incomplete formation of the crystalline phases in the synthesized SSOD. In addition, the broad hump at around 2θ = 25 has clearly been depicted; this is a characteristic peak for amorphous silica sodalite indicating the successful synthesis of SSOD. This result is similar to the XRD pattern for SSOD described by Koike et al. [38]. 

#### 3.1.1. Surface Morphologies of SSOD Crystals

Figure 2 depicts the morphology of the SSOD crystals. The images reveal thin sheet-like shapes arranged together; these are typical shapes of silica sodalite synthesized via topotactic conversion. Similar results were observed by Koike et al. [38] and Ntshangase et al. [20].

#### 3.1.2. Surface Functional Groups of Sodalite Crystals via FT-IR and AFM Analysis

Figure 3 depicts the SSOD FTIR spectrum. The peak at 757 cm^−1^ indicates the stretching vibration of Si-O of the synthesized nanoparticle. A similar observation was reported by Mehmood et al. [39]. The peak at 1100 cm^−1^ is caused by the vibration attributed to the Si-O-S asymmetric stretching mode. It was noted that at around 3300 cm^−1^ there is an absence of a broad peak that belongs to O-H. This could be attributed to the dehydration that occurred during the calcination of SSOD crystals [38].

### 3.2. Characterization of Fabricated Membranes

#### 3.2.1. Surface Morphologies of Fabricated Membranes

Figure 4a,b present the SEM images of the surface, and the cross-sectional area of the pure polysulfone (0 wt.% SSOD/PSF) membrane, respectively. In Figure 4a, the top surface is observed to show clear and uniform morphological features. Figure 4b reveals the typical asymmetric nature of PSF, showing a dense and thin top layer; however, they possess larger pores at the bottom [40]. The structural changes observed on the surface of the polysulfone matrix membrane could be due to the nanoparticles incorporated. Figure 4c,d depict the PSF membrane loaded with 5 wt.% SSOD, and the cross-section PSF membrane loaded with 5 wt.% SSOD, respectively. When comparing the SEM image in Figure 4c with the pure PSF membrane in Figure 4a, more visible pores could be observed on the 5 wt.% SSOD/PSF compared to the pure PSF membrane. This is attributable to the loading of nanoparticles forming aggregates and creating pores on the surface of the membrane. Figure 4d shows visible typical plate-like shapes on the cross-section of the 5 wt.% SSOD/PSF membrane. This resembles the observation made in a study conducted by Koike et al. [38]. Figure 4e presents the morphologies of the top surface of the 10 wt.%SSOD/PSF. The SEM images showed a decrease in the membrane pores as the silica sodalite loading increased from 5 wt.% (Figure 4c) to 10 wt.% (Figure 4e). A similar observation was reported by Alruwaili et al. [41]. This could be a result of the dominance of micro-void creation when nanoparticles embedded in the polymer matrix have been significantly increased [42]. The membrane becomes denser at a higher concentration of nanoparticles, which corroborates the results of Mukherjee and De [43]. Figure 4f depicts the cross-section of 10 wt.% SSOD/PSF membrane. The membrane pores on the cross-section are blocked by the plate-like shapes which are due to the silica sodalite loaded on the polysulfone matrix. Furthermore, the cross-section of the 10 wt.% SSOD/PSF membrane (Figure 4f) shows more plate-like shapes compared to the 5 wt.% SSOD/PSF membrane. This is expected since it was loaded with a higher concentration of SSOD nanoparticles.

#### 3.2.2. Physico-Chemical Properties of Fabricated Membranes

Figure 5 presents the FT-IR spectra of pure polysulfone (0 wt.% SSOD), 5 wt.% SSOD, and 10% wt. SSOD loaded PSF matrix membranes. As a comparison with pure polysulfone, the composite membrane (5% wt. SSOD/PSF) did not show any change in the FTIR spectra; moreover, increasing nanoparticles loading to 10 wt.% also does not have an effect on the spectra. The peaks present on the spectra are assignable to the pure polysulfone membrane. The peak at 1105 cm^−1^ belongs to a saturated C-C single bond. The 1151 cm^−1^ peak confirms the presence of the stretch symmetric C-SO_2_-C group [44]. The sharp and strong peak at 1242 cm^−1^ is due to the presence of a stretching ether (C-O-C) group. A medium peak at 1488 cm^−1^ corresponds to the vibrational stretch of the CH3-C-CH3 bond. Two peaks at 1506 cm^−1^ and 1587 cm^−1^ show a stretch of C=C bond in the aromatic ring this is consistent with the literature [25,44].

#### 3.2.3. Surface Roughness of Fabricated Membranes

The surface roughness was measured using AFM. Figure 6 shows the 3D AFM images together with roughness values. The roughness measurements are used to assess the surface characteristics of the membranes. When there is increased surface roughness, adhesive forces on the surface of the membranes are influenced and become stronger [45]. The adhesive forces are accountable for particle adhesion onto the surface of the membrane, which causes rougher membranes to be easily prone to fouling [46]. The results in Figure 6a showed that the 0 wt.% SSOD/PSF membrane had the maximum average roughness (Ra), and root mean square roughness (Rms) of 163.86 nm and 130.58 nm, respectively. This could be a result of polysulfone hydrophobicity. The roughness values decreased when membranes were embedded with SSOD nanoparticles. However, in Figure 6c, 10 wt.% SSOD loading showed higher surface roughness (Ra = 30.84 nm) compared to the 5 wt.% SSOD-infused PSF (Ra = 19.39 nm) in Figure 6b. This may be due to the slow exchange rate between solvent and polysulfone during phase inversion. The process causes membrane nodules, which result in a rougher surface [23].

#### 3.2.4. Hydrophilicity Measurement of Fabricated Membranes

The membrane’s wettability was determined through the use of contact angle measurements. This is an angle created by a liquid when it is in contact with a solid [28]. For the determination of hydrophilicity or hydrophobicity of the membranes, contact angle measurements were taken, as depicted in Figure 7 [27]. The membrane wetting tendency becomes significant when the contact angle is smaller, hence the affinity between the liquid (water) and solid (membrane) becomes higher. The pure polysulfone measured the maximum contact angle (83.81°) of all the membranes. This may be due to the hydrophobicity of the membrane [47]. When 5 wt.% SSOD nanoparticles were incorporated into the polysulfone matrix, a decrease in contact angle from 83.81° (pure PSF) to 80.94° was observed, showing an improvement in hydrophilicity of the membrane. In addition, a further increase in SSOD loading to 10 wt.% improved the hydrophilic nature of the membrane, since there was a reduction in the contact angle from 80.94° to 76.68°.

#### 3.2.5. Mechanical Properties of Fabricated Membranes

Figure 8a illustrates Young’s modulus for all the membranes and Figure 8b depicts the ultimate tensile strength (UTS) results for all the membranes in this study. The results obtained were used to evaluate the mechanical strength of the fabricated membranes with varied SSOD nanoparticle loading. The results showed that Young’s modulus and ultimate tensile strength (UTS) increased with increasing SSOD loading. The 5 wt.% SSOD/PSF showed an increase in both Young’s modulus and UTS (6.4 and 5 MPa, respectively) in comparison with pure PSF membrane (5.4 and 4.4 MPa, respectively). With a further increase in SSOD loading from 5 wt.% to 10 wt.%, the composite membrane showed an increase in mechanical strength. The increase in both Young’s modulus and UTS indicates that the mechanical strength of the nanocomposite membrane has been enhanced upon adding SSOD nanoparticles [22]. Based on Figure 8, it could be concluded that the highest mechanical strength of the membrane was obtained at SSOD loading of 10 wt.%. It is possible, though not shown in this study, that further addition of the SSOD beyond 10 wt.% might result in the loss of mechanical strength by the membrane. 

### 3.3. Evaluation of the Membranes’ Performance during Phenol-Containing Wastewater Treatment 

The pure water flux and the phenol-containing water flux for membranes having different SSOD loadings at the varied pressure are depicted in Figure 9. It can be seen that fabricated MMM membranes (5 wt.% SSOD/PSF and 10 wt.% SSOD/PSF) possess the highest permeation flux when compared to the pure membrane (0 wt.% SSOD/PSF). The highest flux exhibited by 10 wt.% SSOD/PSF could be linked to the hydrophilic surface as revealed by the contact angle value. It has been found in a previous investigation conducted by Chunjin et al. [48] and Maphutha et al. [49] that hydrophilic membranes resist fouling and as a result have a higher permeation flux. The pure water flux for all the membranes is noticeably higher than phenol-containing water flux; this is due to competitive sorption between phenol and water molecules [31]. Increasing the pressure for both Figure 9a,b resulted in higher permeation flux, since pressure is the driving force for permeation [31].

Figure 10 depicts the rejection performance of the fabricated membranes. Polysulfone with no SSOD loading showed the highest rejection at the lowest pressure of 4 bar compared to the composite membranes. This could be as a result of the SSOD nanoparticles’ agglomeration at specific locations, which affects the phenol rejection in composite membranes. However, a slight increase in the percentage rejection of phenol (64.75%) was noted when the SSOD concentration increased from 5 wt.% to 10 wt.%. When the feed pressure rises from 4 bar to 7 bar, a decrease in the rejection of phenol was noted. This could be ascribed to the increased driving force that forces phenol molecules through the membrane pores [50]. Though the rejection performance of the pure PSF membrane (93.55%) is higher when compared to the membrane loaded with SSOD, membranes loaded with SSOD displayed enhanced permeation flux and mechanical strength when compared to those of the pure Psf membrane. 

Figure 11 depicts the flux recovery ratio of pure Psf membrane and SSOD-loaded membranes. It can be observed that the membrane loaded with 5 wt.% of SSOD was able to recover its initial flux by 87% while flux recovery ratio for the pure Psf membrane (0 wt.%) was about 73% of its initial flux. This is an indication that loading the membrane with SSOD improved the flux recovery ratio of the membrane. However, when the loading increased from 5 wt.% to 10 wt.%, the FRR reduced from 87% to 65%. This obvious reduction in the recovery rate for the membrane loaded with 10 wt.% SSOD could be as a result of poor dispersion of SSOD nanoparticles, forming agglomerates. These agglomerates formed defects on the membrane surface, entrapping the foulant molecules, and thus an effective cleaning process is suggested [20]. 

The membrane fouling is associated with both the reversible and irreversible phenol removal. The reversible fouling is caused by the loose deposition of phenol onto the surface of the membrane after filtration; however, the firmly attached phenol molecules result in irreversible fouling [51]. In addition, the reversible membrane fouling is often removed by flushing with water which occurs during backwashing [52]. Figure 12 depicts the three fouling parameters, namely: total (R_t_), reversible (R_r_), and irreversible (R_ir_) flux decline ratios. As illustrated in Figure 12, Rir of the pure Psf membrane (0 wt.% SSOD/Psf) is about 20.62%, and 57.14% for the 5 wt.% SSOD/Psf and 10 wt.% SSOD/Psf membranes, respectively. The R_ir_ for the 0 wt.% SSOD/Psf is about 4.44% more than the total fouling R_t_, indicating that a lower amount of phenol molecules is irreversibly adsorbed to the surface of the pure Psf membrane. It can be observed that the R_ir_ of the 5 wt.% SSOD/Psf and 10 wt.% SSOD/Psf decreased to 27.78% and 15%, respectively. Obviously, incorporation of the SSOD nanoparticles contributed to the reduction in the irreversible attachment of the phenol molecules. Furthermore, the value of R_r_ of membranes increased with increasing loading of SSOD nanoparticles, indicating that the addition of SSOD nanoparticles resulted in the reversible attachment of phenol molecules. A similar observation was reported by Shen et al. [51]. In the current study, the surface of the 10 wt.% SSOD/Psf membrane is much rougher when compared to that of the 5 wt.% SSOD/Psf, causing the phenol molecules to be embedded within the valley and ridges of the membranes. The findings of this study indicate that fabricated membranes with loaded SSOD could increase phenol-fouling resistance of the membrane during a filtration process.

### 3.4. Conclusions

The following conclusions could be drawn from this study:SSOD nanoparticles are successfully synthesized via topotactic conversion as confirmed by the XRD and SEM results, which is similar to what has been reported in the literature.SEM images showed that incorporation of the SSOD nanoparticles changed the structure of the membrane.The increased concentration of SSOD in the polymer matrix decreased the porosity of the membranes as depicted in SEM images.The pure water flux for all the membranes is noticeably higher than phenol-containing water flux, this is due to competitive sorption between phenol and water molecules.Increasing the concentration of nanoparticles in a polymer of mixed matrix membrane in this study enhanced the hydrophilicity of the membrane, mechanical strength, and permeation flux. The 10 wt.% SSOD/Psf membrane displayed highest flux.Membranes without SSOD particles displayed the highest phenol rejection of 99.55%. However, 10 wt.% SSOD/Psf showed the highest mechanical strengthNonetheless, the permeation flux was increased with increasing nanoparticles loading, implying that nanoparticles cause stronger attraction of water molecules.Findings also showed that the SSOD-loaded membranes displayed an improved phenol fouling resistance during a filtration process.For an improved performance by enhancing the surface area-to-volume ratio of the membrane, a hollow fiber configuration could be considered [53].

The results obtained in this study could be a platform to enhance the SSOD-infused membranes for the treatment of phenol-containing wastewater, in order to maintain the permissible level of phenol in potable water and wastewater, according to World Health Organization’s regulations.

## Figures and Tables

**Figure 1 membranes-12-00800-f001:**
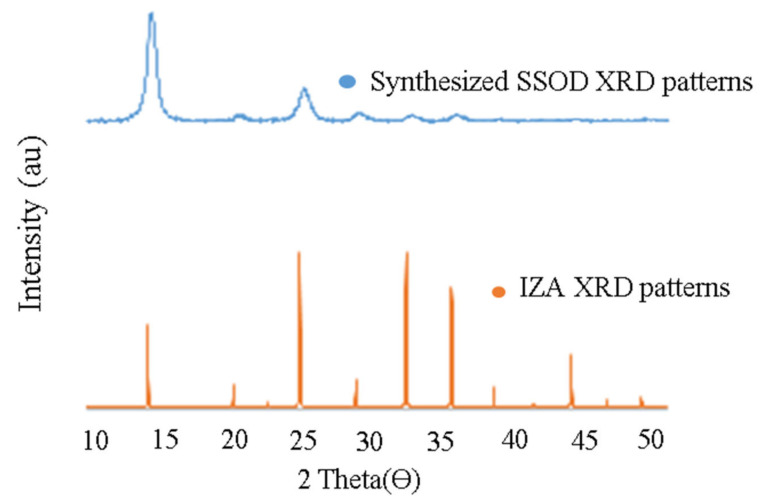
XRD patterns for synthesized SSOD together with reference XRD patterns obtained from the IZA website (Available at: http://www.iza-structure.org/IZA-SC/pow_plot.php, accessed on 13 August 2020).

**Figure 2 membranes-12-00800-f002:**
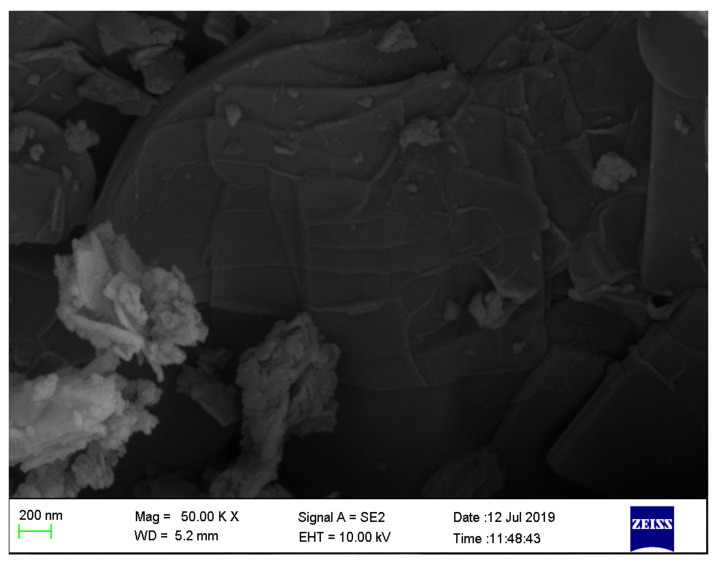
SEM image of silica sodalite.

**Figure 3 membranes-12-00800-f003:**
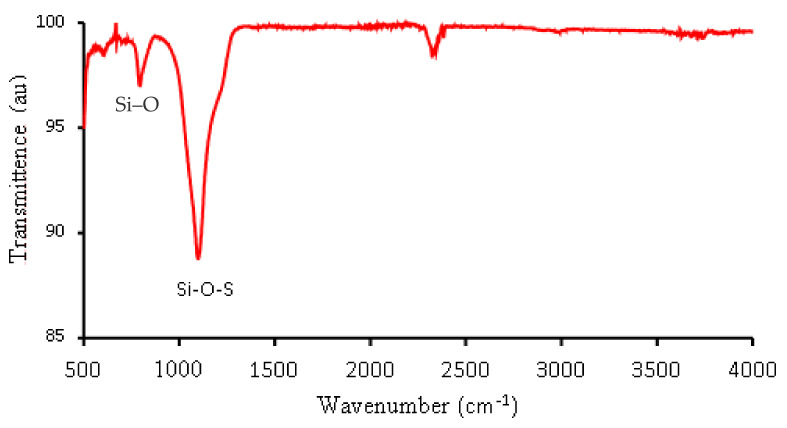
FTIR spectrum SSOD nanoparticles.

**Figure 4 membranes-12-00800-f004:**
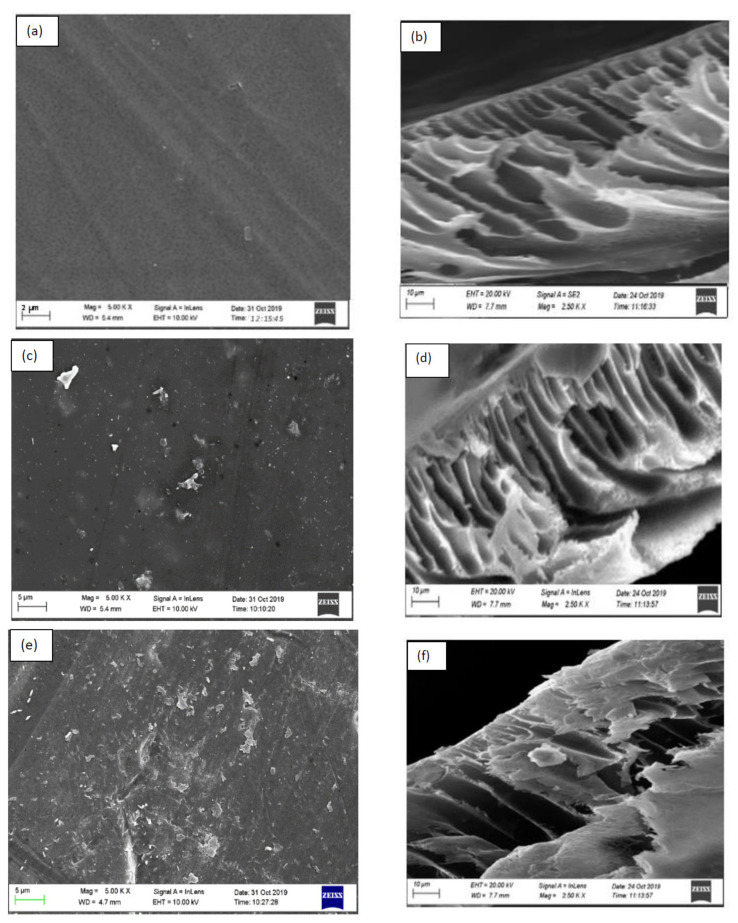
SEM images of (**a**) Top surface of 0 wt.% SSOD/PSF (**b**) Cross-section of 0 wt.% SSOD/PSF, (**c**) Top surface of 5 wt.% SSOD/PSF (**d**) Cross-section of 5 wt.% SSOD/PSF and (**e**) Top surface of 10 wt.% SSOD/PSF (**f**) Cross-section of 10 wt.% SSOD/PSF.

**Figure 5 membranes-12-00800-f005:**
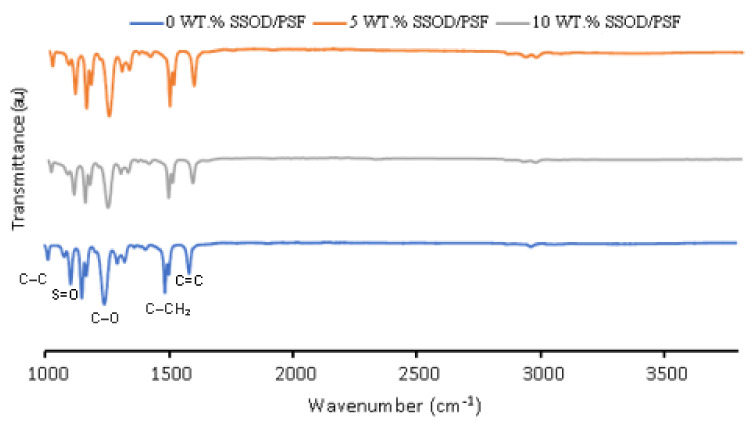
FTIR spectra for the fabricated membranes.

**Figure 6 membranes-12-00800-f006:**
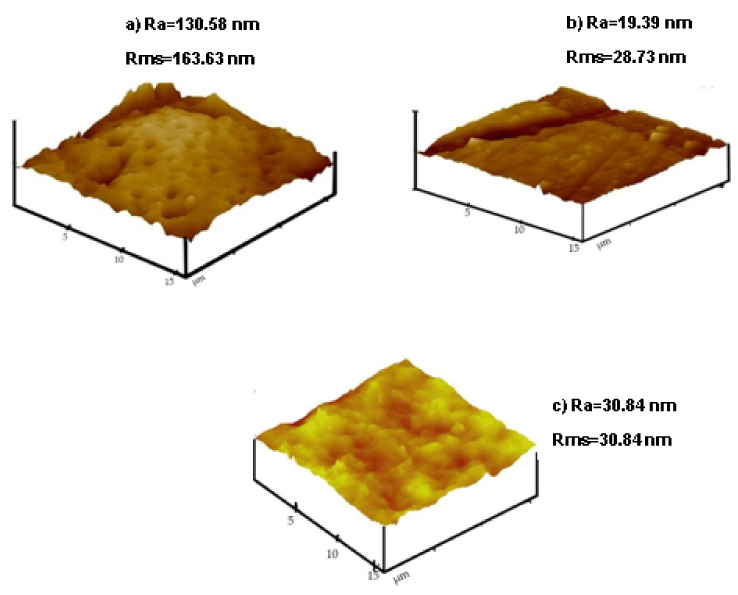
AFM images with surface roughness values (**a**) 0 wt.% SSOD/PSF, (**b**) 5 wt.% SSOD/PSF and (**c**) 10 wt.% SSOD/PSF.

**Figure 7 membranes-12-00800-f007:**
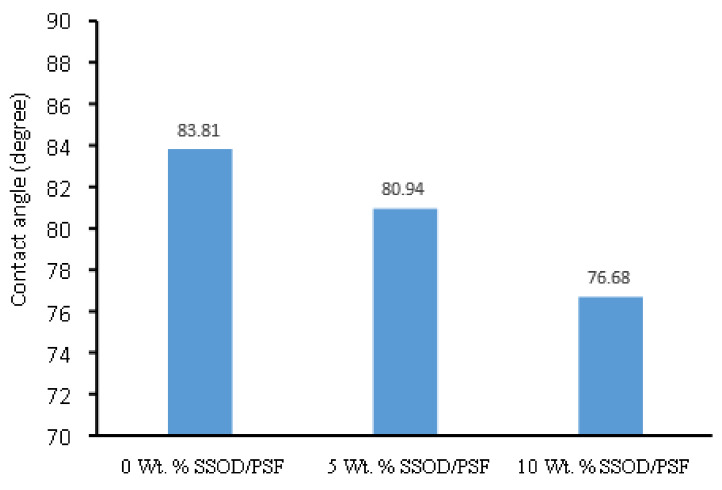
Contact angle measurements.

**Figure 8 membranes-12-00800-f008:**
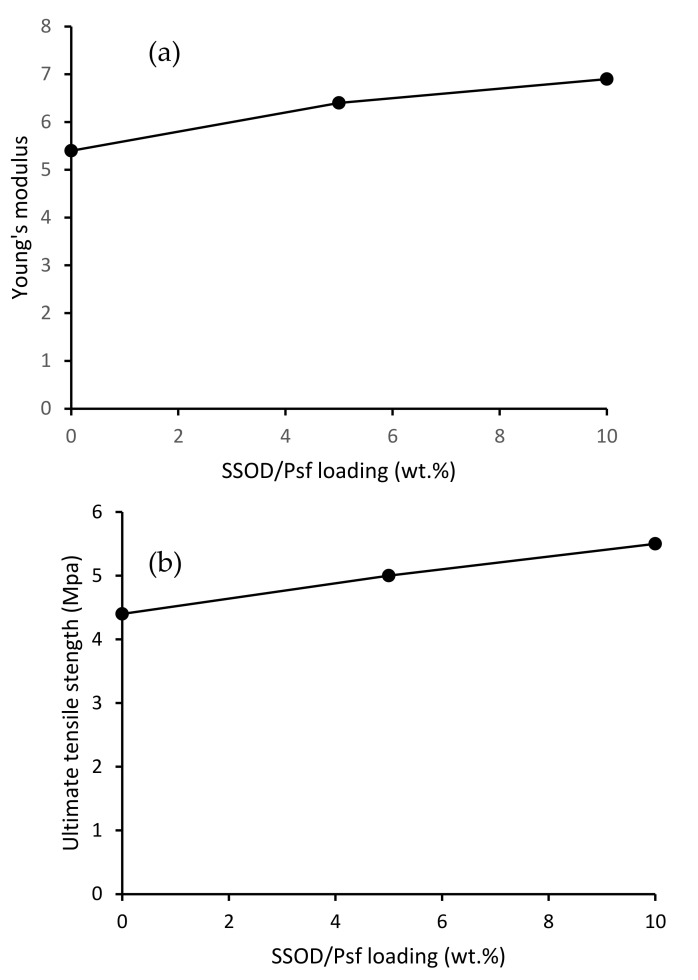
Mechanical properties (**a**) Young’s modulus and (**b**) Ultimate tensile strength.

**Figure 9 membranes-12-00800-f009:**
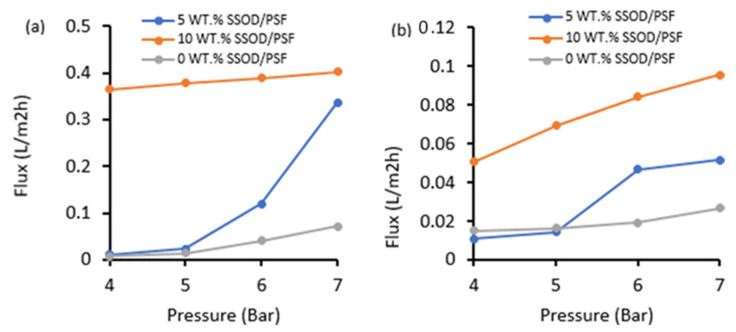
(**a**) Pure water flux (**b**) Phenol-containing water flux.

**Figure 10 membranes-12-00800-f010:**
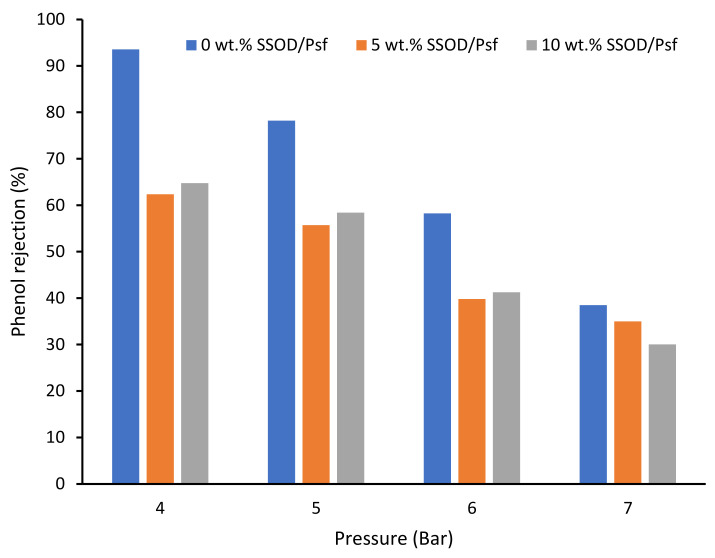
Phenol rejection of membranes at different pressure.

**Figure 11 membranes-12-00800-f011:**
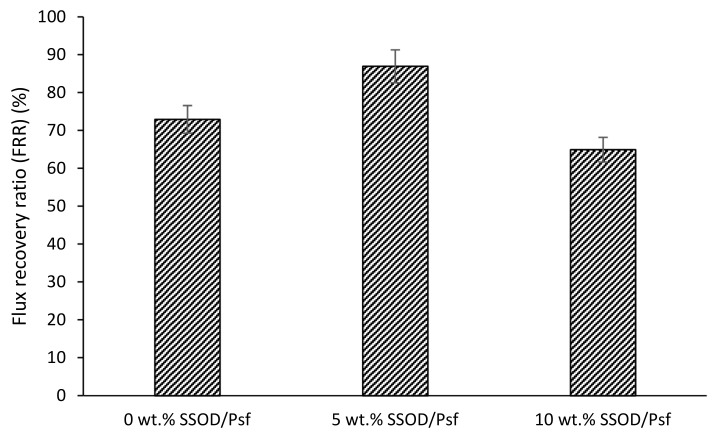
Flux recovery ratio of SSOD/Psf (0–10 wt.%) membranes. Experimental conditions: temperature: 25 °C; Pressure: 4 bar.

**Figure 12 membranes-12-00800-f012:**
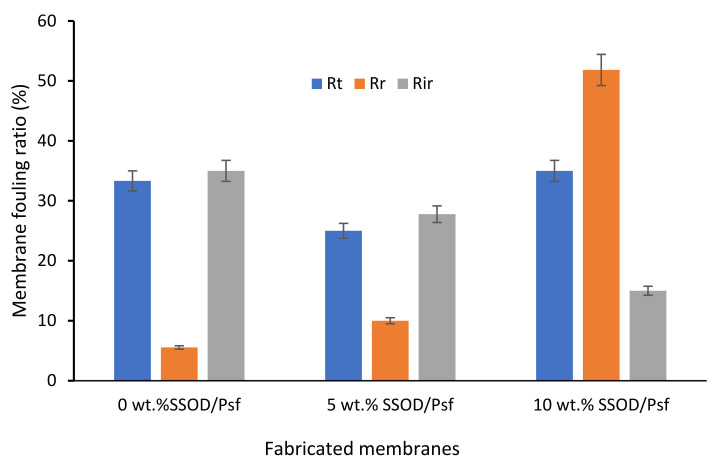
Total fouling ratio (R_t_), irreversible ratio (R_ir_), and reversible fouling ratio (R_r_), of the fabricated membranes during filtration at 4 bar.

**Table 1 membranes-12-00800-t001:** Equilibrium water content and porosity of fabricated membranes.

Membrane	EWC (%)	Porosity (%)
0 wt.% SSOD/Psf	55.01	40.45 ± 1.9
5 wt.% SSOD/Psf	64.05	55.40 ± 2.3
10 wt.% SSOD/Psf	73.24	61.09 ± 0.7

## Data Availability

Not applicable.

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
