# Peer review of "Effect of Silica Sodalite Loading on SOD/PSF Membranes during Treatment of Phenol-Containing Wastewater"

_membranes, 2022, doi:10.3390/membranes12080800_

Round 1
Reviewer 1 Report
Comments to authors:
The authors submitted the manuscript entitled "Effect of silica sodalite loading on SOD/PSF membranes during treatment of phenol-containing wastewater" to be able to publish at Membranes.
The authors prepare hybrid membranes through silica sodalite loading on SOD/PSF to be able to treat the wastewater containing phenol. In fact, the work is very interesting but the authors need to be presented the experiment and results in a very clear way.
The manuscript still needs to be revised and improved in order to be able for publication consideration. So that I cannot recommend the publication of the manuscript in its actual state.A major revisionis recommended.
Some comments are below:
- In the abstract, no need to explain how to prepare the solutions. The final concentration passed through the membranes is sufficient. The details should be presented in the experimental section.
- The authors mentioned that in the abstract, the pores decreased with increasing SSOD. So, in this case, the flux will be reduced. I suggested that the following references to guide authors:
Journal of the Taiwan Institute of Chemical Engineers 129 (2021) 350!360, https://doi.org/10.1016/j.jtice.2021.09.022
Journal of Taibah University for Science
https://doi.org/10.1080/16583655.2021.1885192
Environmental Technology & Innovation 25 (2022) 102210
https://doi.org/10.1016/j.eti.2021.102210
journal of materials research and technology 2022;18:2310e2319
https://doi.org/10.1016/j.jmrt.2022.03.099
Separation and Purification Technology 234 (2020) 116088
https://doi.org/10.1016/j.seppur.2019.116088
Separation and Purification Technology 223 (2019) 17-23
https://doi.org/10.1016/j.seppur.2019.04.057
Chinese Journal of Chemical Engineering (2021), doi: https://doi.org/10.1016/j.cjche.2021.09.019
- The authors need to present the physicochemical properties such as porosity and water uptake.
- The authors should present the Rt, Rr, and Rir Resistance values. Also, they need to provide the FRR Flux recovery ratio.

Author Response
Responses are contained in the attached document. The authors are grateful to the reviewer for these comments. They are useful in improving the scientific quality of the work.
Michael.

Reviewer 2 Report
This article is found interesting and could be useful in membrane based technology. However , I have the following suggestions to improve the quality of manuscript.
1. The diffraction planes observed should be labeled properly in figure 2.
2. The surface morphology is being present without the scale bar and instrumental information. Therefore, it should be revised .
3. FTIR analysis need revision as it is being present without the proper labeling of functional groups.
4. The equation used for average roughness (Ra) and root mean square roughness (Rms) should be added for the better understanding. In this regard, authors can take help from the following article equation/article .
doi.org/10.1016/j.inoche.2021.108606
5. It would be better if the authors can improve the figure 6 with proper scale.
6. Equation used to find the selectivity and percentage rejections of the membrane is being presented without the reference. Authors can help from the following article
7. doi.org/10.1007/s41779-019-00383-x for proper citation of equation.
8. It would better if the authors can modify the interdiction based on the recent challenges and problems regarding the membrane technology for example as stated in following article https://doi.org/10.3390/membranes12070646
Author Response

(The authors gave the same response as above.)

Reviewer 3 Report
For a better article, please revise and add the followed comments.
1. Adding the line between data points is recommended to show the trend clearly in Figure 8.
2. Adding all data of rejection rate from 4 to 7 bar is recommended to show the effect of pressure on phenol rejection clearly in Figure 10.
3. Suggest the optimal SSOD loading rate on PSF membrane for phenol-containing wastewater treatment in results and conclusions.
4. Rejection rate of SSOD loading membrane is much lower than that of pure PSF membrane. What is advantage of your SSOD loading membrane?
5. What is your originality compared with the previous results of Ntshangase et al. [19] and Ngobeni et al. [20]?
Author Response

(The authors gave the same response as above.)

Round 2
Reviewer 1 Report
The authors answer all comments in a reasonable way and modified the manuscript as suggested. So, I recommend the acceptance of the manuscript as it is.
Reviewer 3 Report
This article was revised well, and suitable to be published in Membrane.